# A Novel Approach to Clustering Accelerometer Data for Application in Passive Predictions of Changes in Depression Severity

**DOI:** 10.3390/s23031585

**Published:** 2023-02-01

**Authors:** Mindy K. Ross, Theja Tulabandhula, Casey C. Bennett, EuGene Baek, Dohyeon Kim, Faraz Hussain, Alexander P. Demos, Emma Ning, Scott A. Langenecker, Olusola Ajilore, Alex D. Leow

**Affiliations:** 1Department of Psychiatry, University of Illinois at Chicago, Chicago, IL 60612, USA; 2Department of Biomedical Engineering, University of Illinois at Chicago, Chicago, IL 60612, USA; 3Department of Information and Decision Sciences, University of Illinois at Chicago, Chicago, IL 60612, USA; 4Department of Intelligence Computing, Hanyang University, Seoul 04763, Republic of Korea; 5Department of Computing, DePaul University, Chicago, IL 60604, USA; 6Department of Psychology, University of Illinois at Chicago, Chicago, IL 60612, USA; 7Department of Psychiatry, University of Utah, Salt Lake City, UT 84112, USA

**Keywords:** mood disorders, accelerometer, mHealth, digital phenotyping

## Abstract

The treatment of mood disorders, which can become a lifelong process, varies widely in efficacy between individuals. Most options to monitor mood rely on subjective self-reports and clinical visits, which can be burdensome and may not portray an accurate representation of what the individual is experiencing. A passive method to monitor mood could be a useful tool for those with these disorders. Some previously proposed models utilized sensors from smartphones and wearables, such as the accelerometer. This study examined a novel approach of processing accelerometer data collected from smartphones only while participants of the open-science branch of the BiAffect study were typing. The data were modeled by von Mises-Fisher distributions and weighted networks to identify clusters relating to different typing positions unique for each participant. Longitudinal features were derived from the clustered data and used in machine learning models to predict clinically relevant changes in depression from clinical and typing measures. Model accuracy was approximately 95%, with 97% area under the ROC curve (AUC). The accelerometer features outperformed the vast majority of clinical and typing features, which suggested that this new approach to analyzing accelerometer data could contribute towards unobtrusive detection of changes in depression severity without the need for clinical input.

## 1. Introduction

Mood disorders, such as major depressive disorder and bipolar disorder, often require treatment that is challenging and sometimes a lifelong process [1,2,3]. Traditional methods to monitor mood and other symptoms of these disorders generally rely on self-reports and infrequent clinical visits, which can be time consuming and expensive for the individual and depict an inaccurate portrayal of daily experiences as a result of factors such as recall bias [4,5,6]. While the term mood can convey different meanings, such as momentary mood, which can fluctuate over the course of the day, clinicians instead use these methods to assess the individual’s mood in the context of a disordered state experienced by those with mental illness. As summarized by Hidalgo-Mazzei et al., researchers have begun to incorporate smart technologies into the development of novel methods to monitor mood disorders due in part to the ubiquity of smartphones and wearable devices in the recent years [7]. These devices, which are usually already integrated into the individual’s daily life, contain sensors that have been shown to be capable of unobtrusively detecting future changes in mood [8,9,10,11,12]. The use of technology to supplement traditional treatment approaches has been appealing due to the ability of extracting information on a more granular level than traditional approaches without the need for active input by the user [5]. The passive measures derived are independent of bias and might be a better reflection of everyday life [13].

Previous studies have analyzed a range of data obtained from smartphones and wearable devices, such as GPS location, phone and app usage patterns, voice and ambient noise, and motion sensor information, summarized in several reviews, such as those by Orsolini et al. and Victory et al. in 2020 [14,15]. Through harnessing these data, patterns in an individual’s life could be analyzed to evaluate potential changes related to mood without interrupting daily activities. Since varying moods can lead to differing activity levels [16,17], data recorded from the accelerometer in these devices have been used in combination with other features to passively monitor and predict mood, summarized by Highland and Zhou [18]. Many studies tended to examine metrics from the accelerometer related to the overall movement of the device (e.g., average displacement) as a proxy for the activity level of the individual [10,19,20,21,22,23]. Information about the orientation of the phone was disregarded, possibly due to the inability to identify the activity and location of the phone with respect to the individual without additional information. However, positional information during times of sedentary activities, such as identifying when the individual is laying down versus sitting or standing during different times of the day, could also contribute to passively tracking mood.

One way to obtain this information is by focusing on accelerometer patterns during smartphone keyboard typing. Smartphone typing, which can be performed while sedentary or active, is a common and frequent activity by most smartphone users. The cognitive processes involved while typing are thought to be influenced by mood, which has been supported in previous studies examining the relationship between typing dynamics and symptoms of mood disorders [20,21,24]. Identifying patterns in accelerometer signals during this specific activity could provide additional information related to individuals’ psychological wellbeing. Restricting recording to only during periods of active engagement with the phone in a known orientation allows us to better understand the diurnal patterns of individuals’ phone use and how diurnal changes are related to mood disorders, while also not draining the phone battery from constantly triggering the accelerometer.

In those with mood disorders, diurnal patterns have been found to be disrupted relative to healthy individuals [25,26]. Fluctuations in mood have been noted throughout the day, with mornings generally characterized by worse mood and overall improvement seen as the day progressed [25]. Continued disruptions in these diurnal mood fluctuation patterns may implicate overall deviations in mood [25]. We suspected that sustained deviations from norms in typing position over time might relate to changes in mood. We hypothesized that individuals tend to type on their smartphones in unique but specific orientations depending on their body position, which would be influenced by the time of day and week. Continuous alterations in these positions might be related to overall changes in depression severity. As the conventional clustering algorithms k-means [27], density-based spatial clustering of applications with noise (DBSCAN) [28], and Gaussian mixture models (GMM) were shown to be insufficient in clustering the accelerometer data to identify the preferred phone orientations for all participants, the objective of this study was to develop a novel approach of processing and analyzing accelerometer data longitudinally and to verify their utility to predict clinically relevant changes in depression severity.

## 2. Materials and Methods

### 2.1. The BiAffect iPhone Open Science Study

The participants were a part of the open science branch of the BiAffect study and downloaded the BiAffect app from the Apple app store onto their personal iPhones without the requirement of being a part of a controlled study. Included participants comprised of a combination of those recruited for a controlled study, which utilized the BiAffect app, as well as citizen scientists, who downloaded the app of their own accord. Initially developed to predict changes in mood and cognition for those with bipolar disorder, the study aims to understand whether patterns passively detected through smartphone typing behaviors can be used to monitor mood disorder symptomatology, which could aid in symptom management without increasing the burden on the individual. The app provides users with a custom keyboard that records typing and accelerometer metadata, as well as active cognitive tasks, mood surveys, and rating scales for users to complete periodically. All data collected are de-identified. The data have been used in previous BiAffect studies that found relationships between smartphone keyboard typing patterns and mood disorders [12,20,21,22,29,30,31,32,33].

Specifically, this app recorded the category of keypresses (alphanumeric, backspace, punctuation, etc.) and timestamp while the person was typing using the customized keyboard but not the actual text. Accelerometer readings were also recorded at 10 Hz during typing sessions. In addition to recording typing events, participants were prompted weekly to report the Patient Health Questionaire 8 (PHQ without the suicidality item), which is a self-report of depression severity [34]. As usage of the BiAffect keyboard was entirely voluntary, participation was not consistent. For comparison, missing values were accounted for via two methods: (1) imputation and (2) filtering out of individuals with missing data. Generally, models using imputation performed a few percentage points higher than filtering in terms of accuracy and area under the ROC curve (AUC), which is consistent with previous results [12,33]. In total, there were 295 individuals available in this dataset when using imputation, but only 100 left when using filtering.

### 2.2. Accelerometer Processing

Accelerometer readings were normalized to gravity and recorded as x/y/z coordinates. Readings were filtered to only include coordinates with a magnitude between 0.95 and 1.05 m/s^2^. This filtering resulted in the inclusion of only typing sessions that occurred while the participant was sedentary (standing, sitting, etc.), which was empirically determined using test data obtained during internal testing in various sedentary and active activities. Data were grouped by week for each participant to account for within-week fluctuations in routine, and the von-Mises Fisher (vMF) distribution was calculated for each group (spherical_kde package, version 0.1.0) [35]. This type of distribution was chosen due to the spherical nature of the accelerometer readings, which resembled the unit sphere following normalization and filtering. The vMF distributions were sampled at 1000 equidistant points across a unit sphere to determine the densities of the distribution at a resolution that captured the shape of the distribution while also being sparse enough to not be too computationally expensive [36].

### 2.3. Clustering

To cluster the accelerometer readings per week, the number of clusters in each distribution was first determined through identifying the number of local maxima in the vMF distributions. We reasoned that the peaks in the distribution corresponded to the locations where the majority of the accelerometer points, as well as, subsequently, the number of clusters, lied. Local maxima were defined as points sampled from the vMF distribution with an associated density larger than the 8 neighboring sampled points, with neighbors defined by the number of directly surrounding equidistant sampled points on the vMF distribution. The local maxima with an associated density below a set threshold were attributed to noise and discarded. The threshold was set according to the second bin value of the vMF distribution’s histogram with fifty bins, since the first bin of the histogram contained the portions of the vMF distribution with minimal to no accelerometer readings. The cluster centers were labeled according to the coordinates of the local maxima on the sampled unit sphere.

The common methods to cluster the accelerometer data we chose to compare to our network graph-based method were spherical k-means, DBSCAN, and GMM. The Scikit-Learn package in Python (version 0.21.3) [37] was used to cluster the accelerometer data for DBSCAN and GMM, and the modification of the Scikit-Learn k-means function by the spherecluster package (version 0.1.7) [38] was used for the spherical k-means method. The number of clusters was determined for the spherical k-means and GMM methods using the vMF distribution-based method described above, and the distance metric used for the DBSCAN method was cosine distance due to the spherical nature of the data. All other parameters were left as default.

For our network graph-based method of clustering (shown in Figure 1), the accelerometer readings were assigned to a cluster using graph distance. First, an adjacency matrix was constructed for the equidistant sphere points sampled from the vMF distribution. The edges were weighted using an average of the density sampled from the neighboring points (vMF_i,j_), shown by Equation (1).
(1)$$w_{\mathrm{ij}}=e^{\frac{-\left({\mathrm{vMF}}_{i}+{\mathrm{vMF}}_{j}\right)}{2}}$$

This weighting was used in place of the distance between the neighboring points, since the points were all equally spaced across the unit sphere. Using the average density between two nodes to weight the edge created a graph in which nodes located in a high-density region of the vMF distribution were close together, while nodes located in low density regions were farther apart.

A network graph was constructed using the weighted adjacency matrix (networkx package, version 2.6.3) [39], and the sampled sphere points were assigned to a cluster using Dijkstra’s shortest path algorithm (illustrated in Figure 2). All accelerometer points were matched to the closest sampled sphere point using a nearest neighbor algorithm (scipy.spatial.cKDTree package, version 1.3.1) [40] and assigned the corresponding cluster label.

All processing was conducted in Python, version 3.7.4 [41], using the pandas package (version 1.2.0) [42,43] and NumPy package (version 1.17.2) [44]. Plots were constructed using the matplotlib package, version 3.5.3 [45].

### 2.4. Modeling

Since the accelerometer was selectively recorded only during periods of typing activity, the readings possible were reduced due to the generally limited phone orientations feasible while typing. With that restriction, information about the orientation of the phone was extracted from processed accelerometer readings for longitudinal analysis. This information was possible to infer from the accelerometer alone, since the data were filtered to only include readings of no accelerations aside from that with respect to gravity. By observing the projection of gravity onto the three axes, the relative orientations of the phones were deduced. Orientation and time-based variables calculated from the clustering of weekly accelerometer readings (Table 1) were used, along with typing features, which captured typing speed and variability, as well as clinical factors developed by Bennett et al. [12], to predict clinically relevant changes in PHQ score (difference of 4 or more) [46] using random forest, gradient boosting, and deep learning neural networks methods. Since the majority of the participants included in the dataset did not experience a clinically relevant change in depression, the imbalances were adjusted using synthetic minority oversampling technique (SMOTE) [47]. Feature rankings were determined by a filter-based feature selection method using a random forest model in the python package Scikit-Learn, based on information gain [48]. Odds ratios were calculated in Excel.

## 3. Results

### 3.1. Clustering

The accelerometer data collected was clustered to identify the predominant orientations each participant held their phone while typing. The clustering methods k-means, DBSCAN, and GMM were tested on the accelerometer data itself and generally performed well, but were found to not reliably cluster the points in the expected way that resembled the associated vMF distribution, as shown through two users’ data in Figure 3. For the majority of the participants data, DBSCAN labeled all of the accelerometer points as belonging to one cluster and so was not deemed a feasible method to cluster the data. The spherical k-means algorithm clustered the data evenly between the identified cluster centers and did not account for the density of points when labeling clusters. As seen in Figure 3A, the spherical k-means method split grouped points into separate clusters in a linear fashion. Additionally, unconnected points were sometimes grouped into one cluster (Figure 3B). GMM performed somewhat similarly overall to spherical k-means but tended to group all sparsely located accelerometer points as belonging to one cluster, rather than grouping those points to the nearest cluster of dense accelerometer points (Figure 3A).

Since the common clustering algorithms did not consistently cluster the accelerometer points across different participants and weekly groupings, we developed a new approach to label the points by cluster. Compared to the conventional clustering methods, clustering of the accelerometer data performed using the respective vMF distribution and network graph was found to be the most effective in correctly identifying the number of clusters and labeling the accelerometer readings to the appropriate cluster, as determined through visual inspection (Figure 3). By creating a network graph that resembled the vMF distribution of the accelerometer points, the clusters were mapped to the sampled points from the vMF distribution using the local maxima and weighted graph as a guide. Cluster labels for the accelerometer points themselves were then transferred from the sampled points of the vMF distribution using the nearest neighbor algorithm. This method accommodated irregular and inconsistent cluster shapes between participants’ data by clustering based on the individual network graph. Unlike with the DBSCAN algorithm, separate clusters were identified within regions of overall higher density of accelerometer points within the distribution. Moreover, clusters were allowed to encompass the entire area of higher density, regardless of cluster shape and distance from the cluster center.

The clusters Identified represented the phone orientations that each participant used each week. For example, in Figure 4, one participant held their phone in multiple orientations throughout the course of a week, ranging from deviations of upright (gray, beige, and green clusters) to facing upwards (orange cluster) to horizontally (teal, yellow, and blue clusters).

### 3.2. Changes in Phone Orientation over Time

The labeled accelerometer data was used to identify consistency or changes in typing orientation over time that were unique to each individual. The predominant cluster label per hour was identified and plotted to analyze shifts in typing orientation over time, shown in Figure 5. Some participants had consistent typing orientations, shown by the zero distance from previous cluster on the plot, while other participants shifted typing orientation regularly throughout the day and weeks, which was visualized by frequent changes in the distance from previous cluster on the plot. Larger distances between consecutive clusters corresponded to a more drastic phone orientation change, while smaller distances corresponded to only slight changes in phone orientation. In Figure 5, participant A shifted typing orientation frequently for the majority of the time using the BiAffect keyboard, but spent one week typing only in one orientation. Participant B, on the other hand, predominantly typed in one orientation over the course of a month.

Accelerometer features were calculated from the clustered data to determine their efficacy in predicting clinically relevant changes in PHQ score when combined with typing and clinical features. Model accuracy ranged from 94 to 95.5% (2% standard deviation), with 96 to 98% AUC, similar to that reported in [33]. Importantly, the accelerometer features calculated from the cluster labels contributed greatly to model predictions, even performing better than some otherwise highly ranked clinical features previously reported in the literature [12,49], as can be seen in Table 2.

To understand the directionality of how these PHQ changes were affected by the features, odds ratios were also calculated. The odds ratios for number of clusters and cluster transitions were 1.26 and 1.2, respectively, which suggested that participants with a larger number of clusters and cluster transitions per week had a higher probability of a clinically relevant change in PHQ.

## 4. Discussion

Understanding patterns in a person’s phone orientation while typing could uncover information about changes in their depression severity. While many studies have primarily examined the movement of the phone or wearable device compiled over all activities in analyses using the accelerometer [10,19,20,21,22,23], we sought to determine how the orientation of the phone specifically during typing related to changes in depression severity instead. By limiting the recording times to only during smartphone typing, we were exploring the utility of passive tracking during activities that individuals generally already do on a daily basis while also reducing the toll on the smartphone’s battery. In this study, we developed a novel method of processing accelerometer data to discover how personalized features related to phone orientation can predict clinically relevant changes in depression.

To identify individualized phone orientation tendencies that each participant preferred, clustering of the accelerometer data was performed on a weekly basis to account for day-to-day fluctuations in participants’ schedules due to work and other activities. After first testing the efficacy of conventional clustering methods, we discovered that these algorithms did not consistently provide adequate identification of the different clusters present for all of the participants. As shown in Figure 3, spherical k-means, DBSCAN, and GMM were applied to the accelerometer data and compared to one another.

Although DBSCAN did not require prior knowledge about the number of clusters in the data unlike the other methods used, the algorithm labeled clusters based on the density of the points. Since the accelerometer points for the vast majority of the participants were not well separated between clusters, all points were labeled as belonging to one large cluster, and no distinctions were made between areas of higher and lower densities within the distribution, shown in Figure 3. This method might have worked better if there were clear distinctions between the phone orientations used by the participants, but the overall spread of the points prevented any clear separations.

The spherical k-means algorithm performed well for many of the participants’ data and could reasonable replicate the vMF distributions overall. However, the method works by partitioning the points into clusters, such that each point is labeled as belonging to the nearest cluster center, independent of its location in the distribution [27]. The clusters formed when using this method follow a circular shape, which was not accurate for every instance in our data. Using this method, points that appeared to lie on the edges of a larger cluster might instead be assigned to another cluster solely due to the distance to each cluster center instead of taking into account the shape of the distribution. Moreover, points that were physically separated were sometimes grouped together due to their distances from the cluster centers (Figure 3B).

GMM was an improvement from the spherical k-means algorithm since the algorithm could handle non-circular cluster shapes, but did not seem to handle the sparse points well likely due to noise in the data, as seen in Figure 3A. The distribution of the accelerometer data did not follow a normal distribution, which might have contributed to the subpar performance of the algorithm on the participants’ data overall.

Since the conventional clustering methods tested did not appropriately identify and label the accelerometer data into clusters for all participants, a new approach was developed, outlined in Figure 1 and Figure 2. At the step of clustering the data, the points had already been modeled by vMF distributions in order to determine the number of clusters in each distribution. We then used this representation of the data to create a customized mapping of each distribution for the clustering to preserve their unique characteristics. The clustering using this method was able to accommodate irregular and inconsistent cluster shapes across data from different participants, as well as take into account the varying densities in the vMF distributions when forming the clusters, shown in Figure 3. These clusters represented the different phone orientations participants used each week and depending on the data could suggest several different corresponding body positions, ranging from standing or sitting upright to lounging or laying down, as shown through one participant’s data in Figure 4.

Accelerometer features were then designed as a proxy for body positioning while typing to provide more information about the individual’s environment in models to predict changes in depression severity (Table 1). It is well known that sleep and behavior can be disrupted during periods of depression [50,51,52]. Therefore, we investigated patterns derived from the clustered accelerometer data over time to reveal information about the state of the individual. Reinersten et al. outlined several studies describing that changes in activity can be indicative of changes in depression [16], so we speculated that individuals who typed inconsistent to their usual typing patterns might show signs of a change in depressed mood relative to their previous state. As seen in the comparison between two participants in Figure 5, the patterns of movement between clusters throughout days and weeks ranged in consistency between participants and within participants over time. One participant typed in one orientation over the course of a month aside from a few instances generally in the mornings and nights in which the shift in phone orientation was drastic due to the large distance between consecutive clusters. On the other hand, the other participant had regular shifts in phone orientation throughout the day for many weeks, with some being minor shifts in phone orientation (small distances between consecutive clusters) and others being major shifts in phone orientation (larger distances between consecutive clusters), which suggested that their body position changed from an upright position to variations of lounging or laying down multiple times throughout the day on a regular basis. The shift to consistent phone orientation in week 4 suggested a change in the participant’s behavior during that time, which might have been the result of a change in depression severity. We derived features to capture this information in order to further investigate the relationship between participants’ chosen phone orientation while typing and changes in depression severity. We suspected that how often and the degree to which phone orientations (and therefore body positions) changed over time would be related to participants’ depression severity, with major shifts being more indicative of a change.

We constructed models that examined typing dynamics and clinical measures to predict clinically relevant changes in depression severity in order to evaluate our newly-derived features. Model accuracy was around 95%, and importantly, we observed that the features derived from clustered accelerometer data were very highly ranked in feature importance (Table 2). This ranking suggested that these features, which captured information about the participants’ position while typing, were just as, if not more, important than demographic and clinical information in predicting whether a participant would have a clinically relevant change in depression severity the following week. Moreover, odds ratios, which inform on the direction of change, suggested that the higher number of clusters (i.e., number of typing orientations per week) and number of cluster transitions (i.e., number of times the participant changed between typing positions per week) resulted in an increased likelihood that the participant would experience a change in depression severity the following week. This finding suggested that participants who chose multiple body positions while typing on their smartphone and changed often between them were more likely to experience a fluctuation in their mood, consistent with previous studies evaluating the relationship between psychomotor disturbances, circadian rhythm disruptions, and depression [53,54].

Conventionally, assessments of mood to evaluate treatment efficacy and symptom management rely heavily on a person’s ability to accurately recall their experiences leading up to clinical visits, which can be sparse or difficult to access [6]. These recollections have been previously noted in the literature to often be subject to recall bias [5,55], making regular and accurate monitoring of symptoms difficult. Our approach to processing and analyzing accelerometer data collected during smartphone typing can help to better understand the signals present in this modality and uncover their relationship to mood. Pending further work, models using this approach could be more advantageous in tracking changes in mood due to the greater reliance on passive measures recorded on personal smartphones already used by the majority of the population. The increased granularity of the recorded data and input that does not necessitate access to medical professionals could benefit many individuals by providing objective supplemental information during clinical visits and potentially serve to identify early signs of changes in mood that otherwise might be recognized too late [15,56]. Additional work to investigate this implementation needs to be conducted, though, to determine the extent of beneficial effects of feedback about depression severity, as well as to identify any potentially harmful side effects due to incorrect predictions of changes in depression severity or other factors [57].

This analysis, however, does not come without limitations. First, as the data analyzed belongs to the open-science branch of the BiAffect study, the demographic and clinical information submitted by the participants is not verified by a psychologist and so might not be as accurate as if the data was obtained through a controlled study. Even though the adherence to using the BiAffect keyboard was less consistent as a result of the nature of the participation, the ease to participate facilitated the recruitment and volume of participants.

Furthermore, for the analysis, we elected to filter the data to only include accelerometer readings when the phone was not accelerating independent of gravity, which likely excluded typing sessions while participants were walking or otherwise moving. This exclusion might have impacted the results, but since we did not have any information from other sensors, we would have been unable to deduce the rationale for the accelerations and incorporate them into the models. Future directions could explore the inclusion of this data and the impact on predictions of depression severity.

## 5. Conclusions

In recent years, focus has been placed on more effective and efficient methods to monitor treatment and progression of mood disorder symptoms in order to relieve the burden on individuals. Passive and unobtrusive measures obtained from smartphones and wearable devices already incorporated into most people’s everyday lives have become targets for models to predict future changes in symptomatology. The accelerometer, which is found in most smart devices, has become a popular choice due in part to the ease of information that can be gathered passively without much infringement upon the privacy of the individual compared to other possible information gathered from these devices. This study developed a novel approach of processing accelerometer data that has the potential to augment predictions of changes in depression severity with less dependence on clinical input by the individual.

## Figures and Tables

**Figure 1 sensors-23-01585-f001:**
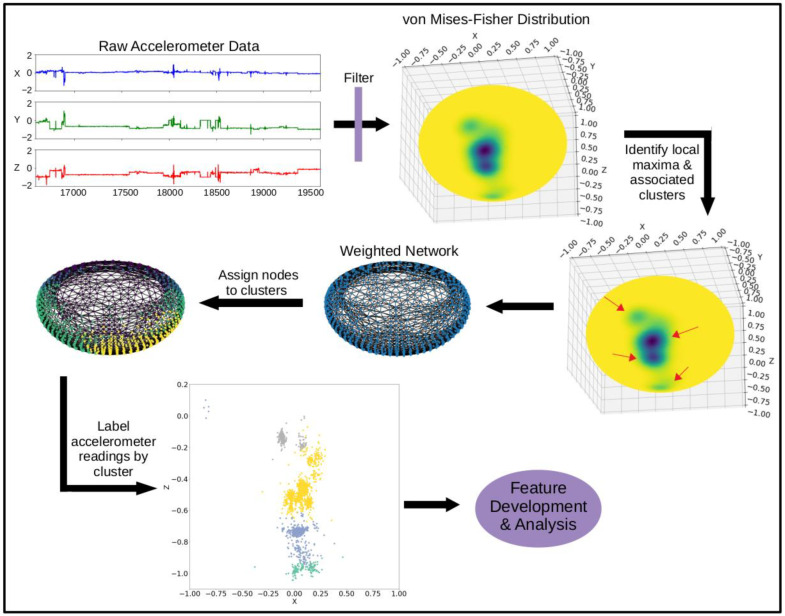
Diagram of steps to cluster accelerometer data.

**Figure 2 sensors-23-01585-f002:**
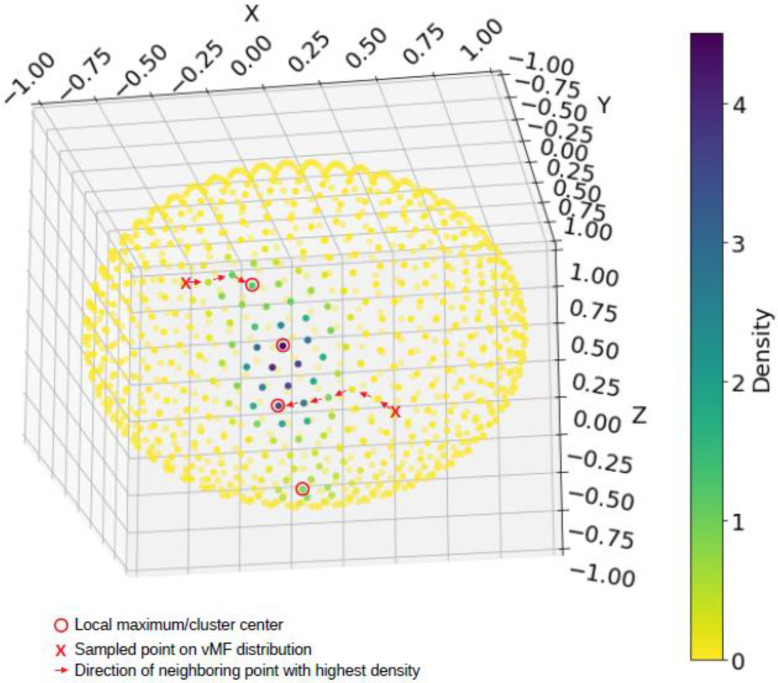
Each of the 1000 equidistant points (that jointly discretized the unit sphere) is assigned to one of the local maxima by following a path of increasing kernal density (i.e., a gradient ascent procedure on the kernal density function), as illustrated using the points marked by an X. This is algorithmically implemented by forming a weighted graph followed by computing the shortest path length between points.

**Figure 3 sensors-23-01585-f003:**
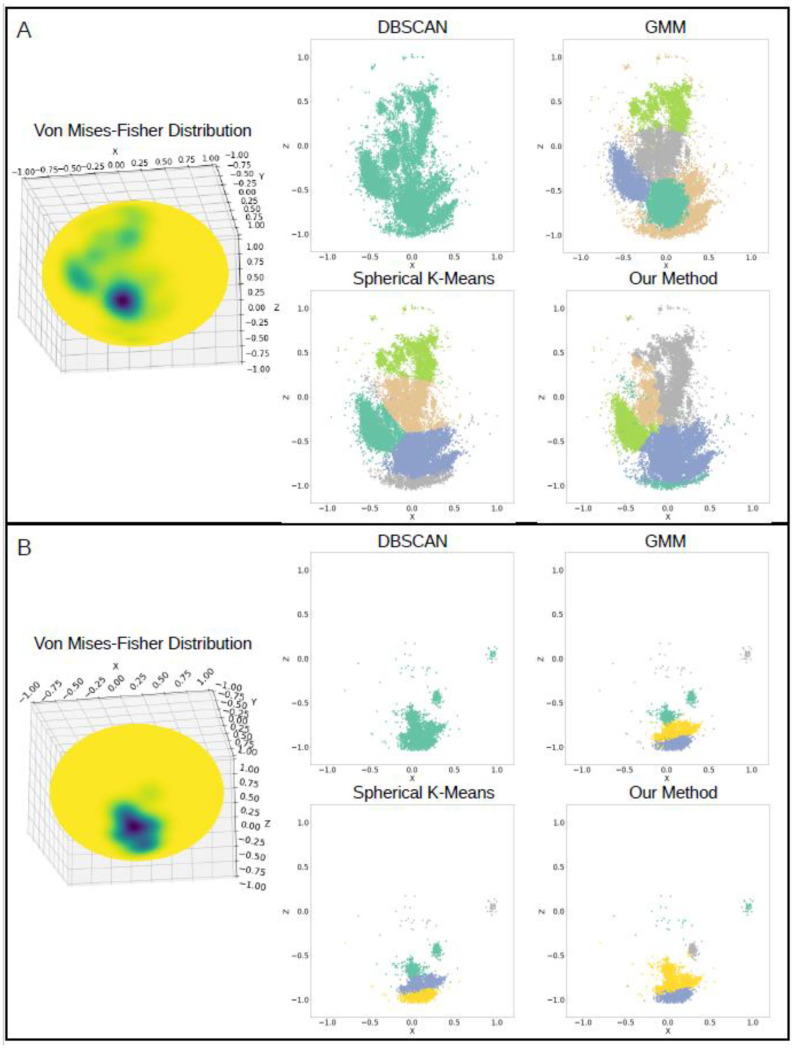
Comparison of conventional clustering methods tested (DBSCAN, spherical k-means, GMM) to our method for two users’ data (**A**,**B**) over a week.

**Figure 4 sensors-23-01585-f004:**
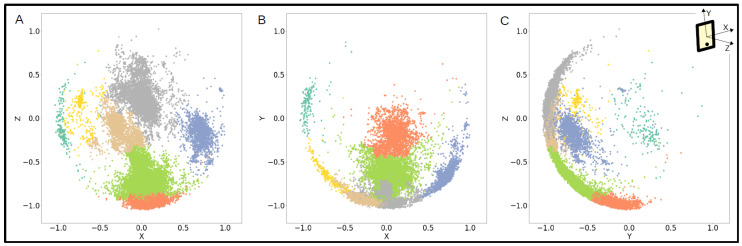
One participant’s accelerometer data over a week labeled by cluster and plotted in the (**A**) xz axes, (**B**) xy axes, and (**C**) yz axes.

**Figure 5 sensors-23-01585-f005:**
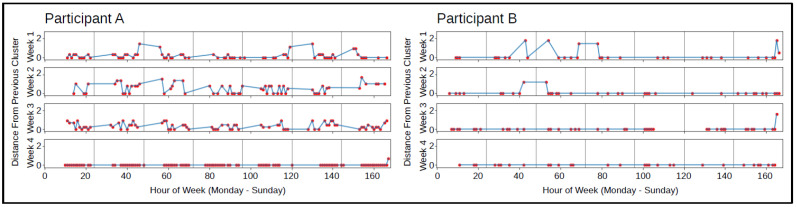
Two participants’ plots of the haversine distance between consecutive cluster centers per hour over the weeks using the BiAffect keyboard.

**Table 1 sensors-23-01585-t001:** Accelerometer features calculated using clustered accelerometer data.

Feature Name	Feature Description
n_clusters	Number of distinct phone typing orientations per week
total_distance_between_clusters	Sum of haversine distances traveled between session cluster labels per week
avg_n_clusters_perSession	Average number of cluster labels of x/y/z readings within each session per week
avg_n_transitions_perSession	Average number of changes between consecutive cluster labels of x/y/z readings within each session per week
n_cluster_transitions	Number of changes between consecutive session cluster labels per week
median_X, median_Y, median_Z	Median x/y/z reading of session’s cluster center per week
sum_X_motion, sum_Y_motion, sum_Z_motion	Sum of differences between consecutive cluster center x/y/z readings of session cluster labels per week
X_motion_sd, Y_motion_sd, Z_motion_sd	Standard deviation of differences between consecutive cluster center x/y/z readings of session cluster labels per week
arc_sum	Three-dimensional rotational motion per week (calculated based on session cluster center x/y/z accelerometer readings)

**Table 2 sensors-23-01585-t002:** Feature rankings based on information gain filter-based method (using a random forest model) [48] with accelerometer features calculated from cluster labels in bold.

Feature Ranking
**median_Y**
**median_X**
Age
**n_clusters**
**median_Z**
BD
**n_cluster_transitions**
**sum_Z_motion**
BD_binary
MDQdiag
phoneSize
Gender
Anxiety
medianPressDur
PTSD
**sum_Y_motion**
**sum_X_motion**
Depression
OCD
percent_upright_night
medianDistCenter
**arc_sum**
**avg_n_transitions_perSession**
ADHD
count_X_horizontal
n_XYZ
**avg_n_clusters_perSession**
**X_motion_sd**
NoneOfTheseDiag
**total_distance_between_clusters**
distToCenterPrevRatioAA
autocorrectRate_wkSD
medianIKD
Avg90PercentileAA
autocorrectRate
AvgVarAB
SubstanceAddictionDisorder
backspaceRate
AvgVarAA
**Z_motion_sd**
medIKD_wkSD
Avg_nBackspace
percent_upright_afternoon
percent_upright_morning
bkspRate_wkSD
Avg_nAutocorrect
AvgVarBB
percent_upright
SeasonalAffectiveDisorder
nKeypresses
**Y_motion_sd**
percent_upright_evening
AvgMedAA
Avg_medPressDuration
AvgMAD_AA
Diag_PreferNotAnswer
AvgMedAB
Avg_nAlphanum
Schizophrenia
AvgMedBB

## Data Availability

The data presented in this study are available upon request from the corresponding author. The data are not publicly available due to privacy concerns.

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
