# Peer review of "A Novel Approach to Clustering Accelerometer Data for Application in Passive Predictions of Changes in Depression Severity"

_sensors, 2023, doi:10.3390/s23031585_

Round 1
Reviewer 1 Report
The authors present a novel methodological approach to tackle a very pressing topic, the association of physical activity and mood states. While I think the approach is very interesting and adds to the current knowledge in general, I think the manuscript should be specified in several aspects:
1) The authors claim to predict mood changes, but their outcome variable are PHQ-change-scores. Since mood and PHQ-scores mirror not the same constructs, the authors should rewrite and sharpen their wording overall and especially their conclusions.
2) related to this (1): I would expect affect and mood to change on a momentary state level and the PHQ-scores to vary on a different time scale. Maybe the authors want to discuss such timing issues in their manuscript, too.
3) also related to (1): The clinical interview is considered gold standard in psychiatry, therefore I would argue to reword the introduction: The onobtrousive passive methods may predict this gold standard, but we do not know how smartphone signals relate to momentary well-being and symptomatology unless we capture the latter ones, e.g., via ecological momentary assessements, see:
https://www.ncbi.nlm.nih.gov/pmc/articles/PMC7430559/
https://psycnet.apa.org/buy/2009-22537-002
4) 2.2 Acc processing: 0.95 - 1.05 - what's the unit here ? "...as determined previously..." - here's a citation missing for the evidence the authors refer to.
5) for 2.3 it would be good to include a graph to make these steps more understandable for readers
6) 2.4 "...clinically relevant changes in PHQ..." => Reference is missing
7) The conclusion is in parts overinterpreting the results in my opinion. As stated above, we do not know if the assessment is a proxi for mood changes since the outcome variable measured and predicted were change scores in the PHQ. Therefore, I would like to encourage the authors to carefully review their wording, especially in the discussion and conclusion sections.
Reviewer 2 Report
Overall, very interesting and impactful work!
Line 24 – “…from the clustered data to..” seems like the to isn’t needed?
Line 26 May want to define AUC
Lines 37-41 Very long run-on sentence – please rework
Lines 65-73 – Seems like this information would be better suited in the methods. Also, do you have any other references that can back up WHY you decided to do while they are typing…you gave great rationale above for using these data and how they are collected but may be good to provide one or two references to why just typing (if available) – if not available, please state that. The information presented with the hypothesis feels as though it should come prior.
These analyses are secondary from the BiAffect study? Please state these are secondary analyses and I’m assuming one of the cited references is about the BiAffect study? Would be helpful to specifically cite which one it is to reference to understand the procedures of that study better.
Is the app phone specific – iPhone and Android both or just one vs the other? If both, do they use the same type of accelerometer in the phone models? Not sure if this could be an issue with reliability/validity/sensitivity?
Is there a reason that the clustering algorithms did not consistently cluster the data points (line 202)?
In the limitations, you state that the filtering of the data reduced the available data/sample – is the sample size also an issue or is 100 more than enough, just more so the available data points?
Also, the data used were on the weekly level – would it be more informative to use the data at a daily level? – This is not a criticism, just out of curiosity and may be useful to state WHY the weekly data was used.
While these analyses and findings are so important and could be hugely impactful, what are some next steps? Make the keyboard/app more widely available/standard on all smartphones? Is it in the works to be integrated into medical record data? Will the processing of the data be available to medical care teams and patients alike - I know it would be incredibly enlightening to me to see how my own mood changes based on my typing and positioning.
